# Stochastic Gradient Riemannian Langevin Dynamics on the Probability Simplex

**Sam Patterson**
Gatsby Computational Neuroscience Unit
University College London
spatterson@gatsby.ucl.ac.uk

**Yee Whye Teh**
Department of Statistics
University of Oxford
y.w.teh@stats.ox.ac.uk

## Abstract

In this paper we investigate the use of Langevin Monte Carlo methods on the probability simplex and propose a new method, Stochastic gradient Riemannian Langevin dynamics, which is simple to implement and can be applied to large scale data. We apply this method to latent Dirichlet allocation in an online mini-batch setting, and demonstrate that it achieves substantial performance improvements over the state of the art online variational Bayesian methods.

## 1   Introduction

In recent years there has been increasing interest in probabilistic models where the latent variables or parameters of interest are discrete probability distributions over $K$ items, i.e. vectors lying in the probability simplex

$$\Delta_K = \{(\pi_1, \ldots, \pi_K) : \pi_k \geq 0, \sum_k \pi_k = 1\} \subset \mathbb{R}^K \tag{1}$$

Important examples include topic models like latent Dirichlet allocation (LDA) [BNJ03], admixture models in genetics like Structure [PSD00], and discrete directed graphical models with a Bayesian prior over the conditional probability tables [Hec99].

Standard approaches to inference over the probability simplex include variational inference [Bea03, WJ08] and Markov chain Monte Carlo methods (MCMC) like Gibbs sampling [GRS96]. In the context of LDA, many methods have been developed, e.g. variational inference [BNJ03], collapsed variational inference [TNW07, AWST09] and collapsed Gibbs sampling [GS04]. With the increasingly large scale document corpora to which LDA and other topic models are applied, there has also been developments of specialised and highly scalable algorithms [NASW09]. Most proposed algorithms are based on a batch learning framework, where the whole document corpus needs to be stored and accessed for every iteration. For very large corpora, this framework can be impractical.

Most recently, [Sat01, HBB10, MHB12] proposed online Bayesian variational inference algorithms (OVB), where on each iteration only a small subset (a mini-batch) of the documents is processed to give a noisy estimate of the gradient, and a stochastic gradient descent algorithm [RM51] is employed to update the parameters of interest. These algorithms have shown impressive results on very large corpora like Wikipedia articles, where it is not even feasible to store the whole dataset in memory. This is achieved by simply fetching the mini-batch articles in an online manner, processing, and then discarding them after the mini-batch.

In this paper, we are interested in developing scalable MCMC algorithms for models defined over the probability simplex. In some scenarios, and particularly in LDA, MCMC algorithms have been shown to work extremely well, and in fact achieve better results *faster* than variational inference on small to medium corpora [GS04, TNW07, AWST09]. However current MCMC methodology

have mostly been in the batch framework which, as argued above, cannot scale to the very large corpora of interest. We will make use of a recently developed MCMC method called stochastic gradient Langevin dynamics (SGLD) [WT11, ABW12] which operates in a similar online mini-batch framework as OVB. Unlike OVB and other stochastic gradient descent algorithms, SGLD is not a gradient descent algorithm. Rather, it is a Hamiltonian MCMC [Nea10] algorithm which will asymptotically produce samples from the posterior distribution. It achieves this by updating parameters according to both the stochastic gradients as well as additional noise which forces it to explore the full posterior instead of simply converging to a MAP configuration.

There are three difficulties that have to be addressed, however, to successfully apply SGLD to LDA and other models defined on probability simplices. Firstly, the probability simplex (1) is compact and has boundaries that has to be accounted for when an update proposes a step that brings the vector outside the simplex. Secondly, the typical Dirichlet priors over the probability simplex place most of its mass close to the boundaries and corners of the simplex. This is particularly the case for LDA and other linguistic models, where probability vectors parameterise distributions over a larger number of words, and it is often desirable to use distributions that place significant mass on only a few words, i.e. we want distributions over $\Delta_K$ which place most of its mass near the boundaries and corners. This also causes a problem as depending on the parameterisation used, the gradient required for Langevin dynamics is inversely proportional to entries in $\pi$ and hence can blow up when components of $\pi$ are close to zero. Finally, again for LDA and other linguistic models, we would like algorithms that work well in high-dimensional simplices.

These considerations lead us to the first contribution of this paper in Section 3, which is an investigation into different ways to parameterise the probability simplex. This section shows that the choice of a good parameterisation is not obvious, and that the use of the Riemannian geometry of the simplex [Ama95, GC11] is important in designing Langevin MCMC algorithms. In particular, we show that an unnormalized parameterisation, using a mirroring trick to remove boundaries, coupled with a natural gradient update, achieves the best mixing performance. In Section 4, we then show that the SGLD algorithm, using this parameterisation and natural gradient updates, performs significantly better than OVB algorithms [HBB10, MHB12]. Section 2 reviews Langevin dynamics, natural gradients and SGLD to setup the framework used in the paper, and Section 6 concludes.

## 2 Review

### 2.1 Langevin dynamics

Suppose we model a data set $\mathbf{x} = x_1, \ldots, x_N$, with a generative model $p(\mathbf{x} \mid \theta) = \prod_{i=1}^{N} p(x_i \mid \theta)$ parameterized by $\theta \in \mathbb{R}^D$ with prior $p(\theta)$ and that our aim is to compute the posterior $p(\theta \mid \mathbf{x})$. Langevin dynamics [Ken90, Nea10] is an MCMC scheme which produces samples from the posterior by means of gradient updates plus Gaussian noise, resulting in a proposal distribution $q(\theta^* \mid \theta)$ as described by Equation 2.

$$\theta^* = \theta + \frac{\epsilon}{2}\left(\nabla_\theta \log p(\theta) + \sum_{i=1}^{N} \nabla_\theta \log p(x_i|\theta)\right) + \zeta, \qquad \zeta \sim N(0, \epsilon I) \qquad (2)$$

The mean of the proposal distribution is in the direction of increasing log posterior due to the gradient, while the added noise will prevent the samples from collapsing to a single (local) maximum. A Metropolis-Hastings correction step is required to correct for discretisation error, with proposals accepted with probability $\min\left(1, \frac{p(\theta^*|\mathbf{x})}{p(\theta|\mathbf{x})}\frac{q(\theta|\theta^*)}{q(\theta^*|\theta)}\right)$ [RS02]. As $\epsilon$ tends to zero, the acceptance ratio tends to one as the Markov chain tends to a stochastic differential equation which has $p(\theta \mid \mathbf{x})$ as its stationary distribution [Ken78].

### 2.2 Riemannian Langevin dynamics

Langevin dynamics has an isotropic proposal distribution leading to slow mixing if the components of $\theta$ have very different scales or if they are highly correlated. Preconditioning can help with this. A recent approach, the Riemann manifold Metropolis adjusted Langevin algorithm [GC11] uses a user chosen matrix $G(\theta)$ to precondition in a locally adaptive manner. We will refer to their algorithm

as Riemannian Langevin dynamics (RLD) in this paper. The Riemannian manifold in question is the family of probability distributions $p(x \mid \theta)$ parameterised by $\theta$, for which the expected Fisher information matrix $\mathcal{I}_\theta$ defines a natural Riemannian metric tensor. In fact any positive definite matrix $G(\theta)$ defines a valid Riemannian manifold and hence we are not restricted to using $G(\theta) = \mathcal{I}_\theta$. This is important in practice as for many models of interest the expected Fisher information is intractable.

As in Langevin dynamics, RLD consists of a Gaussian proposal $q(\theta^* \mid \theta)$, along with a Metropolis-Hastings correction step. The proposal distribution can be written as

$$\theta^* = \theta + \frac{\epsilon}{2}\mu(\theta) + G^{-\frac{1}{2}}(\theta)\zeta, \qquad \zeta \sim N(0, \epsilon I) \tag{3}$$

where the $j^{th}$ component of $\mu(\theta)$ is given by

$$\mu(\theta)_j = \left( G^{-1}(\theta) \left( \nabla_\theta \log p(\theta) + \sum_{i=1}^{N} \nabla_\theta \log p(x_i|\theta) \right) \right)_j - 2\sum_{k=1}^{D} \left( G^{-1}(\theta) \frac{\partial G(\theta)}{\partial \theta_k} G^{-1}(\theta) \right)_{jk}$$

$$+ \sum_{k=1}^{D} \left( G^{-1}(\theta) \right)_{jk} \operatorname{Tr}\left( G^{-1}(\theta) \frac{\partial G(\theta)}{\partial \theta_k} \right) \tag{4}$$

The first term in Equation 4 is now the natural gradient of the log posterior. Whereas the standard gradient gives the direction of steepest ascent in Euclidean space, the natural gradient gives the direction of steepest descent taking into account the geometry implied by $G(\theta)$. The remaining terms in Equation 4 describe how the curvature of the manifold defined by $G(\theta)$ changes for small changes in $\theta$. The Gaussian noise in Equation 3 also takes the geometry of the manifold into account, having scale defined by $G^{-\frac{1}{2}}(\theta)$.

## 2.3 Stochastic gradient Riemannian Langevin dynamics

In the Langevin dynamics and RLD algorithms, the proposal distribution requires calculation of the gradient of the log likelihood w.r.t. $\theta$, which means processing all $N$ items in the data set. For large data sets this is infeasible, and even for small data sets it may not be the most efficient use of computation. The stochastic gradient Langevin dynamics (SGLD) algorithm [WT11] replaces the calculation of the gradient over the full data set, with a stochastic approximation based on a subset of data. Specifically at iteration $t$ we sample $n$ data items indexed by $D_t$, uniformly from the full data set and replace the exact gradient in Equation 2 with the approximation

$$\nabla_\theta \log p(\mathbf{x} \mid \theta) \approx \frac{N}{|D_t|} \sum_{i \in D_t} \nabla_\theta \log p(x_i|\theta) \tag{5}$$

Also, SGLD does not use a Metropolis-Hastings correction step, as calculating the acceptance probability would require use of the full data set, hence defeating the purpose of the stochastic gradient approximation. Convergence to the posterior is still guaranteed as long as decaying step sizes satisfying $\sum_{t=1}^{\infty} \epsilon_t = \infty, \sum_{t=1}^{\infty} \epsilon_t^2 < \infty$ are used.

In this paper we combine the use of a preconditioning matrix $G(\theta)$ as in RLD with this stochastic gradient approximation, by replacing the exact gradient in Equation 4 with the approximation from Equation 5. The resulting algorithm, stochastic gradient Riemannian Langevin dynamics (SGRLD), avoids the slow mixing problems of Langevin dynamics, while still being applicable in a large scale online setting due to its use of stochastic gradients and lack of Metropolis-Hastings correction steps.

## 3 Riemannian Langevin dynamics on the probability simplex

In this section, we investigate the issues which arise when applying Langevin Monte Carlo methods, specifically the Langevin dynamics and Riemannian Langevin dynamics algorithms, to models whose parameters lie on the probability simplex. In these experiments, a Metropolis-Hastings correction step was used. Consider the simplest possible model: a $K$ dimensional probability vector $\pi$ with Dirichlet prior $p(\pi) \propto \prod_k^K \pi_k^{\alpha_k-1}$, and data $\mathbf{x} = x_1, \ldots, x_N$ with $p(x_i = k \mid \pi) = \pi_k$. This results in a Dirchlet posterior $p(\pi \mid \mathbf{x}) \propto \prod_k^K \pi_k^{n_k+\alpha_k-1}$, where $n_k = \sum_{i=1}^{N} \delta(x_i = k)$. In

| Parameterisation | Reduced-Mean | Reduced-Natural | Expanded-Mean | Expanded-Natural |
|---|---|---|---|---|
| $\theta$ | $\theta_k = \pi_k$ | $\theta_k = \log\frac{\pi_k}{1-\sum_{k=1}^{K-1}\pi_k}$ | $\pi_k = \frac{\|\theta_k\|}{\sum_{k=1}\|\theta_k\|}$ | $\pi_k = \frac{e^{\theta_k}}{\sum_{k=1}e^{\theta_k}}$ |
| $\nabla_\theta \log p(\theta\|\mathbf{x})$ | $\frac{n+\alpha}{\theta} - \mathbf{1}\frac{n_K+\alpha-1}{\pi_K}$ | $n+\alpha - (n_.+K\alpha)\pi$ | $\frac{n+\alpha-1}{\theta} - \frac{n_.}{\theta_.} - \mathbf{1}$ | $n+\alpha - n_.\pi - e^\theta$ |
| $G(\theta)$ | $n_.\left(\mathrm{diag}(\theta)^{-1} + \frac{1}{1-\sum_k\theta_k}\mathbf{11}^T\right)$ | $\frac{1}{n_.}\left(\mathrm{diag}(\pi)-\pi\pi^T\right)$ | $\mathrm{diag}(\theta)^{-1}$ | $\mathrm{diag}(e^\theta)$ |
| $G^{-1}(\theta)$ | $\frac{1}{n_.}\left(\mathrm{diag}(\theta)-\theta\theta^T\right)$ | $n_.\left(\mathrm{diag}(\pi)^{-1}+\frac{1}{1-\sum_k\pi_k}\mathbf{11}^T\right)$ | $\mathrm{diag}(\theta)$ | $\mathrm{diag}(e^{-\theta})$ |
| $\sum_{k=1}^D\left(G^{-1}\frac{\partial G}{\partial\theta_k}G^{-1}\right)_{jk}$ | $K\theta_j - 1$ | $\frac{1}{\pi_j^2} - \frac{K-1}{(1-\sum_k\pi_k)^2}$ | $-1$ | $e^{-\theta_j}$ |
| $\sum_{k=1}^D\left(G^{-1}(\theta)\right)_{jk}\mathrm{Tr}\left(G^{-1}(\theta)\frac{\partial G}{\partial\theta_k}\right)$ | $K\theta_j - 1$ | $\frac{1}{\pi_j^2} - \frac{K-1}{(1-\sum_k\pi_k)^2}$ | $-1$ | $e^{-\theta_j}$ |

Table 1: Parameterisation Details

our experiments we use a sparse, symmetric prior with $\alpha_k = 0.1\,\forall k$, and sparse count data, setting $K = 10$ and $n_1 = 90$, $n_2 = n_3 = 5$ and the remaining $n_k$ to zero. This is to replicate the sparse nature of the posterior in many models of interest. The qualitative conclusions we draw are not sensitive to the precise choice of hyperparameters and data here.

There are various possible ways to parameterise the probability simplex, and the performance of Langevin Monte Carlo depends strongly on the choice of parameterisation. We consider both the mean and natural parameter spaces, and in each of these we try both a reduced ($K-1$ dimensional) and expanded ($K$ dimensional) parameterisation, with details as follows.

**Reduced-Mean:** in the mean parameter space, the most obvious approach is to set $\theta = \pi$ directly, but there are two problems with this. Though $\pi$ has $K$ components, it must lie on the simplex, a $K-1$ dimensional space. Running Langevin dynamics or RLD on the full $K$ dimensional parameterisation will result in proposals that are off the simplex with probability one. We can incorporate the constraint that $\sum_{k=1}^K \pi_k = 1$ by using the first $K-1$ components as the parameter $\theta$, and setting $\pi_K = 1 - \sum_{k=1}^{K-1}\pi_k$. Note however that the proposals can still violate the boundary constraint $0 < \pi_k < 1$, and this is particularly problematic when the posterior has mass close to the boundaries.

**Expanded-Mean:** we can simplify boundary considerations using a redundant parameterisation. We take as our parameter $\theta \in \mathbb{R}_+^K$ with prior a product of independent Gamma($\alpha_k$, 1) distributions, $p(\theta) \propto \prod_{k=1}^K \theta_k^{\alpha_k-1}e^{-\theta_k}$. $\pi$ is then given by $\pi_k = \frac{\theta_k}{\sum_k\theta_k}$ and so the prior on $\pi$ is still Dirichlet($\alpha$). The boundary conditions $0 < \theta_k$ can be handled by simply taking the absolute value of the proposed $\theta^*$. This is equivalent to letting $\theta$ take values in the whole of $\mathbb{R}^K$, with prior given by Gammas mirrored at 0, $p(\theta) \propto \prod_{k=1}^K |\theta_k|^{\alpha_k-1}e^{-|\theta_k|}$, and $\pi_k = \frac{|\theta_k|}{\sum_k|\theta_k|}$, which again results in a Dirichlet($\alpha$) prior on $\pi$. This approach allows us to bypass boundary issues altogether.

**Reduced-Natural:** in the natural parameter space, the reduced parameterisation takes the form $\pi_k = \frac{e^{\theta_k}}{1+\sum_{k=1}^{K-1}e^{\theta_k}}$ for $k = 1,\ldots,K-1$. The prior on $\theta$ can be obtained from the Dirichlet($\alpha$) prior on $\pi$ using a change of variables. There are no boundary constraints as the range of $\theta_k$ is $\mathbb{R}$.

**Expanded-Natural:** finally the expanded-natural parameterisation takes the form $\pi_k = \frac{e^{\theta_k}}{\sum_{k=1}^K e^{\theta_k}}$ for $k = 1,\ldots,K$. As in the expanded-mean parameterisation, we use a product of Gamma priors, in this case for $e^{\theta_k}$, so that the prior for $\pi$ remains Dirichlet($\alpha$).

For all parameterisations, we run both Langevin dynamics and RLD. When applying RLD, we must choose a metric $G(\theta)$. For the reduced parameterisations, we can use the expected Fisher information matrix, but the redundancy in the full parameterisations means that this matrix has rank $K-1$ and hence is not invertible. For these parameterisations we use the expected Fisher information matrix for a Gamma/Poisson model, which is equivalent to the Dirichlet/Multinomial apart from the fact that the total number of data items is considered to be random as well.

The details for each parameterisation are summarised in Table 1. In all cases we are interested in sampling from the posterior distribution on $\pi$, while $\theta$ is the specific parameterisation being used. For the mean parameterisations, the $\theta^{-1}$ term in the gradient of the log-posterior means that for components of $\theta$ which are close to zero, the proposal distribution for Langevin dynamics (Equation 2) has a large mean, resulting in unstable proposals with a small acceptance probability. Due to the form of $G(\theta)^{-1}$, the same argument holds for the RLD proposal distribution for the natural parameterisations. This leaves us with three possible combinations, RLD on the expanded-mean parameterisation and Langevin dynamics on each of the natural parameterisations.

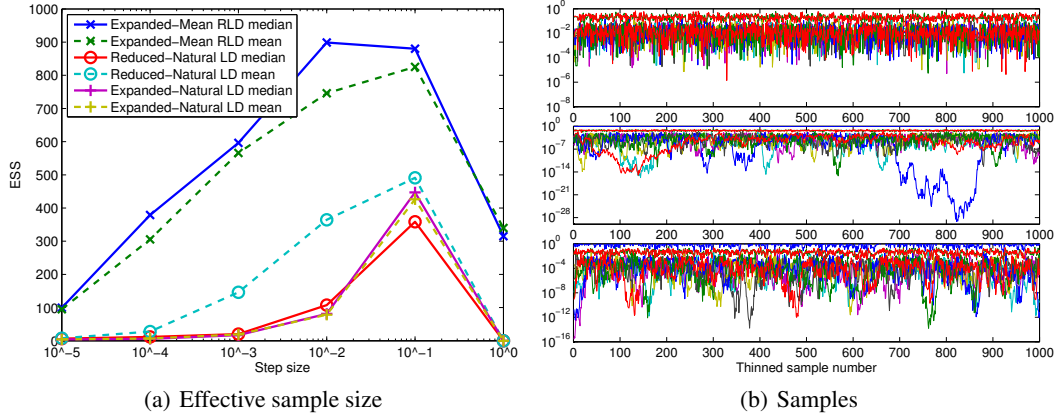

(a) Effective sample size          (b) Samples

Figure 1: Effective sample size and samples. Burn-in iterations is 10,000; thinning factor 100.

To investigate their relative performances we run a small experiment, producing 110,000 samples from each of the three remaining parameterisations, discarding 10,000 burn-in samples and thinning the remaining samples by a factor of 100. For the resulting 1000 thinned samples of $\theta$, we calculate the corresponding samples of $\pi$, and compute the effective sample size for each component of $\pi$. This was done for a range of step sizes $\epsilon$, and the mean and median effective sample sizes for the components of $\pi$ is shown in Figure 1(a).

Figure 1(b) shows the samples from each sampler at their optimal step size of 0.1. The samples from Langevin dynamics on both natural parameterisations display higher auto-correlation than the RLD samples produced using the expanded-mean parameterisation, as would be expected from their lower effective sample sizes. In addition to the increased effective sample size, the expanded-mean parameterisation RLD sampler has the advantage that it is computationally efficient as $G(\theta)$ is a diagonal matrix. Hence it is this algorithm that we use when applying these techniques to latent Dirichlet allocation in Section 4.

## 4 Applying Riemannian Langevin dynamics to latent Dirichlet allocation

Latent Dirichlet Allocation (LDA) [BNJ03] is a hierarchical Bayesian model, most frequently used to model topics arising in collections of text documents. The model consists of $K$ topics $\pi_k$, which are distributions over the words in the collection, drawn from a symmetric Dirichlet prior with hyper-parameter $\beta$. A document $d$ is then modelled by a mixture of topics, with mixing proportion $\eta_d$, drawn from a symmetric Dirichlet prior with hyper-parameter $\alpha$. The model corresponds to a generative process where documents are produced by drawing a topic assignment $z_{di}$ i.i.d. from $\eta_d$ for each word $w_{di}$ in document $d$, and then drawing the word $w_{di}$ from the corresponding topic $\pi_{z_{di}}$.

We integrate out $\eta$ analytically, resulting in the semi-collapsed distribution:

$$p(w, z, \pi \mid \alpha, \beta) = \prod_{d=1}^{D} \frac{\Gamma(K\alpha)}{\Gamma(K\alpha + n_{d\cdot\cdot})} \prod_{k=1}^{K} \frac{\Gamma(\alpha + n_{dk\cdot})}{\Gamma(\alpha)} \prod_{k=1}^{K} \frac{\Gamma(W\beta)}{\Gamma(\beta)^W} \prod_{w=1}^{W} \pi_{kw}^{\beta + n_{\cdot kw} - 1} \quad (6)$$

where as in [TNW07], $n_{dkw} = \sum_{i=1}^{N_d} \delta(w_{di} = w, z_{di} = k)$ and $\cdot$ denotes summation over the corresponding index. Conditional on $\pi$, the documents are i.i.d., and we can factorise Equation 6

$$p(w, z, \pi \mid \alpha, \beta) = p(\pi \mid \beta) \prod_{d=1}^{D} p(w_d, z_d \mid \alpha, \pi) \quad (7)$$

where

$$p(w_d, z_d, \mid \alpha, \pi) = \prod_{k=1}^{K} \frac{\Gamma(\alpha + n_{dk\cdot})}{\Gamma(\alpha)} \prod_{w=1}^{W} \pi_{kw}^{n_{dkw}} \quad (8)$$

### 4.1 Stochastic gradient Riemannian Langevin dynamics for LDA

As we would like to apply these techniques to large document collections, we use the stochastic gradient version of the Riemannian Langevin dynamics algorithm, as detailed in Section 2.3. Following the investigation in Section 3 we use the expanded-mean parameterisation. For each of the $K$ topics $\pi_k$, we introduce a $W$-dimensional unnormalised parameter $\theta_k$ with an independent Gamma prior $p(\theta_k) \propto \prod_{w=1}^{W} \theta_{kw}^{\beta_w-1} e^{-\theta_{kw}}$ and set $\pi_{kw} = \frac{\theta_{kw}}{\sum_w \theta_{kw}}$, for $w = 1, \ldots, W$. We use the mirroring idea as well. The metric $G(\theta)$ is then the diagonal matrix $G(\theta) = \text{diag} \left( \theta_{11}, \ldots, \theta_{1W}, \ldots, \theta_{K1}, \ldots, \theta_{KW} \right)^{-1}$.

The algorithm runs on mini-batches of documents: at time $t$ it receives a mini-batch of documents indexed by $D_t$, drawn at random from the full corpus $D$. The stochastic gradient of the log posterior of $\theta$ on $D_t$ is shown in Equation 9.

$$\frac{\partial \log p(\theta \mid w, \beta, \alpha)}{\partial \theta_{kw}} \approx \frac{\beta - 1}{\theta_{kw}} - 1 + \frac{|D|}{|D_t|} \sum_{d \in D_t} \mathbb{E}_{z_d | w_d, \theta, \alpha} \left[ \frac{n_{dkw}}{\theta_{kw}} - \frac{n_{dk\cdot}}{\theta_{k\cdot}} \right] \tag{9}$$

For this choice of $\theta$ and $G(\theta)$, we use Equations 3, 4 to give the SGRLD update for $\theta$,

$$\theta_{kw}^* = \left| \theta_{kw} + \frac{\epsilon}{2} \left( \beta - \theta_{kw} + \frac{|D|}{|D_t|} \sum_{d \in D_t} \mathbb{E}_{z_d | w_d, \theta, \alpha} \left[ n_{dkw} - \pi_{kw} n_{dk\cdot} \right] \right) + (\theta_{kw})^{\frac{1}{2}} \zeta_{kw} \right| \tag{10}$$

where $\zeta_{kw} \sim \mathcal{N}(0, \epsilon)$. Note that the $\beta - 1$ term in Equation 9 has been replaced with $\beta$ in Equation 10 as the $-1$ cancels with the curvature terms as detailed in Table 1. As discussed in Section 3, we reflect moves across the boundary $0 < \theta_{kw}$ by taking the absolute value of the proposed update.

Comparing Equation 9 to the gradient for the simple model from Section 3, the observed counts $n_k$ for the simple model have been replaced with the expectation of the latent topic assignment counts $n_{dkw}$. To calculate this expectation we use Gibbs sampling on the topic assignments in each document separately, using the conditional distributions

$$p(z_{di} = k \mid w_d, \theta, \alpha) = \frac{\left( \alpha + n_{dk\cdot}^{\backslash i} \right) \theta_{kw_{di}}}{\sum_k \left( \alpha + n_{dk\cdot}^{\backslash i} \right) \theta_{kw_{di}}} \tag{11}$$

where $\backslash i$ represents a count excluding the topic assignment variable we are updating.

## 5 Experiments

We investigate the performance of SGRLD, with no Metropolis-Hastings correction step, on two real-world data sets. We compare it to two online variational Bayesian algorithms developed for latent Dirichlet allocation: online variational Bayes (OVB) [HBB10] and hybrid stochastic variational-Gibbs (HSVG) [MHB12]. The difference between these two methods is the form of variational assumption made. OVB assumes a mean-field variational posterior, $q(\eta_{1:D}, z_{1:D}, \pi_{1:K}) = \prod_d q(\eta_d) \prod_{d,i} q(z_{di}) \prod_k q(\pi_k)$, in particular this means topic assignment variables within the same document are assumed to be independent, when in reality they will be strongly coupled. In contrast HSVG collapses $\eta_d$ analytically and uses a variational posterior of the form $q(z_{1:D}, \pi_{1:K}) = \prod_d q(z_d) \prod_k q(\pi_k)$, which allows dependence within the components of $z_d$. This more complicated posterior requires Gibbs sampling in the variational update step for $z_d$, and we combined the code for OVB [HBB10], with the Gibbs sampling routine from our SGRLD code to implement HSVG.

### 5.1 Evaluation Method

The predictive performance of the algorithms can be measured by looking at the probability they assign to unseen data. A metric frequently used for this purpose is perplexity, the exponentiated cross entropy between the trained model probability distribution and the empirical distribution of the test data. For a held-out document $w_d$ and a training set $\mathcal{W}$, the perplexity is given by

$$\text{perp}(w_d \mid \mathcal{W}, \alpha, \beta) = \exp \left\{ -\frac{\sum_{i=1}^{n_{d\cdot\cdot}} \log p(w_{di} \mid \mathcal{W}, \alpha, \beta)}{n_{d\cdot\cdot}} \right\}. \tag{12}$$

This requires calculating $p(w_{di} \mid \mathcal{W}, \alpha, \beta)$, which is done by marginalising out the parameters $\eta_d, \pi_1, \ldots, \pi_K$ and topic assignments $z_d$, to give

$$p(w_{di} \mid \mathcal{W}, \alpha, \beta) = \mathbb{E}_{\eta_d, \pi} \left[ \sum_k \eta_{dk} \pi_{kw_{di}} \right] \tag{13}$$

We use a document completion approach [WMSM09], partitioning the test document $w_d$ into two sets of words, $w_d^{\text{train}}, w_d^{\text{test}}$ and using $w_d^{\text{train}}$ to estimate $\eta_d$ for the test document, then calculating the perplexity on $w_d^{\text{test}}$ using this estimate.

To calculate the perplexity for SGRLD, we integrate $\eta$ analytically, so Equation 13 is replaced by

$$p(w_{di} \mid w_d^{\text{train}}, \mathcal{W}, \alpha, \beta) = \mathbb{E}_{\pi \mid \mathcal{W}, \beta} \left[ \mathbb{E}_{z_d^{\text{train}} \mid \pi, \alpha} \left[ \sum_k \hat{\eta}_{dk} \pi_{kw_{di}} \right] \right] \tag{14}$$

where

$$\hat{\eta}_{dk} := p(z_{di}^{\text{test}} = k \mid z_d^{\text{train}}, \alpha) = \frac{n_{dk\cdot}^{\text{train}} + \alpha}{n_{d\cdot\cdot}^{\text{train}} + K\alpha}. \tag{15}$$

We estimate these expectations using the samples we obtain for $\pi$ from the Markov chain produced by SGRLD, and samples for $z_d^{\text{train}}$ produced by Gibbs sampling the topic assignments on $w_d^{\text{train}}$.

For OVB and HSVG, we estimate Equation 13 by replacing the true posterior $p(\eta, \beta)$ with $q(\eta, \beta)$.

$$p(w_{di} \mid \mathcal{W}, \alpha, \beta) = \mathbb{E}_{p(\eta_d, \pi \mid \mathcal{W}, \alpha, \beta)} \left[ \sum_k \eta_{dk} \pi_{kw_{di}} \right] \approx \sum_k \mathbb{E}_{q(\eta_d)} \left[ \eta_{dk} \right] \mathbb{E}_{q(\pi_k)} \left[ \pi_{kw_{di}} \right] \tag{16}$$

We estimate the perplexity directly rather than use a variational bound [HBB10] so that we can compare results of the variational algorithms to those of SGRLD.

## 5.2 Results on NIPS corpus

The first experiment was carried out on the collection of NIPS papers from 1988-2003 [GCPT07]. This corpus contains 2483 documents, which is small enough to run all three algorithms in batch mode and compare their performance to that of collapsed Gibbs sampling on the full collection. Each document was split 80/20 into training and test sets, the training portion of all 2483 documents were used in each update step, and the perplexity was calculated on the test portion of all documents. Hyper-parameters $\alpha$ and $\beta$ were both fixed to 0.01, and 50 topics were used. A step-size schedule of the form $\epsilon_t = (a * (1 + \frac{t}{b}))^{-c}$ was used. Perplexities were estimated for a range of step size parameters, and for 1, 5 and 10 document updates per topic parameter update. For OVB the document updates are fixed point iterations of $q(z_d)$ while for HSVG and SGRLD they are Gibbs updates of $z_d$, the first half of which were discarded as burn-in. These numbers of document updates were chosen as previous investigation of the performance of HSVG for varying numbers of Gibbs updates has shown that 6-10 updates are sufficient [MHB12] to achieve good performance.

Figure 2(a) shows the lowest perplexities achieved along with the corresponding parameter settings. As expected, CGS achieves the lowest perplexities. It is surprising that HSVG performs slightly worse than OVB on this data set. As it uses a less restricted variational distribution it should perform at least as well. SGRLD improves on the performance of OVB and HSVG, but does not match the performance of Gibbs sampling.

## 5.3 Results on Wikipedia corpus

The algorithms' performances in an online scenario was assessed on a set of articles downloaded at random from Wikipedia, as in [HBB10]. The vocabulary used is again as per [HBB10]; it is not created from the Wikipedia data set, instead it is taken from the top 10,000 words in Project Gutenburg texts, excluding all words of less than three characters. This results in vocabulary size $W$ of approximately 8000 words. 150,000 documents from Wikipedia were used in total, in mini-batches of 50 documents each. The perplexities were estimated using the methods discussed in

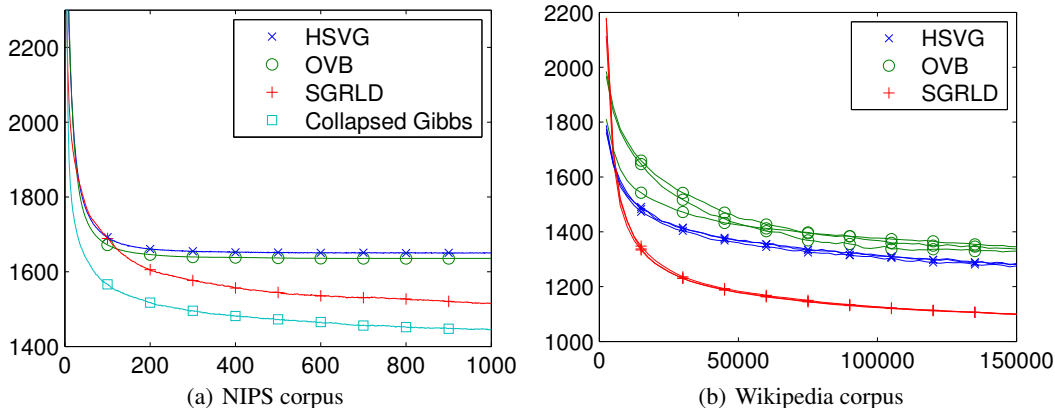

Figure 2: Test-set perplexities on NIPS and Wikipedia corpora.

Section 5.1 on a separate holdout set of 1000 documents, split 90/10 training/test. As the corpus size is large, collapsed Gibbs sampling was not run on this data set.

For each algorithm a grid-search was run on the hyper-parameters, step-size parameters, and number of Gibbs sampling sweeps / variational fixed point iterations per $\pi$ update. The lowest three perplexities attained for each algorithm are shown in Figure 2(b). Corresponding parameters are given in the supplementary material. HSVG achieves better performance than OVB, as expected. The performance of SGRLD is a substantial improvement on both the variational algorithms.

## 6 Discussion

We have explored the issues involved in applying Langevin Monte Carlo techniques to a constrained parameter space such as the probability simplex, and developed a novel online sampling algorithm which addresses those issues. Using an expanded parametrisation with a reflection trick for negative proposals removed the need to deal with boundary constraints, and using the Riemannian geometry of the parameter space dealt with the problem of parameters with differing scales.

Applying the method to Latent Dirichlet Allocation on two data sets produced state of the art predictive performance for the same computational budget as competing methods, demonstrating that full Bayesian inference using MCMC can be practically applied to models of interest, even when the data set is large. Python code for our method is available at `http://www.stats.ox.ac.uk/~teh/sgrld.html`.

Due to the widespread use of models defined on the probability simplex, we believe the methods developed here for Langevin dynamics on the probability simplex will find further uses beyond latent Dirichlet allocation and stochastic gradient Monte Carlo methods. A drawback of SGLD algorithms is the need for decreasing step sizes; it would be interesting to investigate adaptive step sizes and the approximation entailed when using fixed step sizes (but see [AKW12] for a recent development).

**Acknowledgements**

We thank the Gatsby Charitable Foundation and EPSRC (grant EP/K009362/1) for generous funding, reviewers and area chair for feedback and support, and [HBB10] for use of their excellent publicly available source code.

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
