[Supplementary Material · sgrld-supp.pdf]

# Stochastic Gradient Riemannian Langevin Dynamics on the Probability Simplex
## —Supplementary Material—

**Sam Patterson**
Gatsby Computational Neuroscience Unit
University College London
spatterson@gatsby.ucl.ac.uk

**Yee Whye Teh**
Department of Statistics
University of Oxford
y.w.teh@stats.ox.ac.uk

| Algorithm | a | b | c | $\alpha$ | $\beta$ | K | Gibbs samples / Fixed-pt iterations |
|---|---|---|---|---|---|---|---|
| OVB | 0.1 | 10000 | 0.7 | 0.01 | 0.01 | 50 | 5 |
| HSVG | 0.1 | 1000 | 0.7 | 0.01 | 0.01 | 50 | 5 |
| SGRLD | 0.05 | 1000 | 0.7 | 0.01 | 0.01 | 50 | 10 |

Table 1: Parameter settings for NIPS experiment.

| Algorithm | a | b | c | $\alpha$ | $\beta$ | K | Gibbs samples / Fixed-pt iterations |
|---|---|---|---|---|---|---|---|
| OVB | 0.01 | 10000 | 0.6 | 0.01 | 0.1 | 100 | 100 |
| OVB | 0.01 | 1000 | 0.6 | 0.01 | 0.1 | 100 | 200 |
| OVB | 0.001 | 10000 | 0.6 | 0.01 | 0.1 | 100 | 100 |
| HSVG | 0.001 | 10000 | 0.8 | 0.01 | 0.001 | 100 | 200 |
| HSVG | 0.001 | 10000 | 0.6 | 0.01 | 0.1 | 100 | 200 |
| HSVG | 0.001 | 10000 | 0.8 | 0.01 | 0.1 | 100 | 200 |
| SGRLD | 0.001 | 10000 | 0.6 | 0.01 | 0.0001 | 100 | 200 |
| SGRLD | 0.001 | 1000 | 0.6 | 0.01 | 0.0001 | 100 | 200 |
| SGRLD | 0.001 | 1000 | 0.6 | 0.01 | 0.1 | 100 | 200 |

Table 2: Parameter settings for Wikipedia experiment.