[Reviews · NeurIPS 2013]

Submitted by Assigned_Reviewer_4

This paper extends the stochastic gradient langevin dynamics by using the Reimannian structure and applies it into probability simplex. The idea appears to be quite interesting. But there are several confusing parts that I don't quite get. Maybe the authors can elaborate those a bit.

* Novelty: The novel part of this paper is that it generalized the stochastic gradient langevin dynamics to probability simplex by using the Reimannian conditioning. However, I have some questions here. It seems to me that this is more like a parameter transformation and I am not quite how a preconditioning matrix helps here. Is preconditioning necessary to achieve better performance? Did you try to use a non-preconditioned version?

also Why do you take the absolute value as in line 198? What does this mean for the posterior sampling? Is there a theory behind this?

* Writing: the writing is generally clear.

* Experiments: The experiments look good. Can you give some intuitions or explanations why the proposed method is better?
Summary: This paper extends the stochastic gradient langevin dynamics by using the Reimannian structure and applies it into probability simplex. The idea appears to be quite interesting. But there are several confusing parts that I don't quite get. Maybe the authors can elaborate those a bit.

Submitted by Assigned_Reviewer_5

The authors introduce a stochastic gradient approximate MCMC algorithm appropriate for parameters living on the probability simplex. They experiment with four different parameterisations (the combinations of redundant vs. not and mean vs natural parameters). A simple Riemannian space is also used. The method is applied to LDA, using the idea of MHB12 to calculate the expectations over the assignments z: Gibbs sampling. Performance on two reasonably large datasets is superior to the current state of the art online variational methods.

Quality. The quality of this work is very good: the ideas are good and are well presented, I like that they show the alternative parameterisations that they experimented rather than just showing the final algorithm, and the experiments while not exhaustive are sufficient to support the validity of the algorithm.

Clarity. Excellent. All required background is given succintly.

Originality. Most of the components involved are known: for example Riemannian Langevin dynamics and using Gibbs sampling in a stochastic gradient type scheme for LDA to get expectations wrt z. The authors do not cite any prior working using the expanded mean parameterisation, but it seems suprising if this has not been used before anywhere.

Significance. Scaling Bayesian hierarchical models is important if they are to be used in the wild. I have actually wondered for a while the "variational" part of online variational methods was really required when stochastic gradient sampling options like Langevin dynamics exist. This paper answers that question in the negative, which is an interesting result for the field.

My only minor criticism of the paper is that the figures are quite hard to read (and I think would be even worse printed): the text needs to be much larger at least, and maybe show fewer combinations of parameters in fig 2b.
Summary: A good paper, should have written it myself.

Submitted by Assigned_Reviewer_6

This paper combines the recently proposed stochastic gradient Langevin dynamics with Riemannian Langevin dynamics. The new sampling algorithm is able to use a locally adaptive preconditioning matrix for a faster convergence rate. It is applied to draw samples in the posterior distribution of the LDA model under the online learning setting, and achieves a better predictive performance than online variational Bayesian methods.

The combination of SGLD with RLD provides a promising direction to improve the application of MCMC methods on large scale data set. It is usually difficult to choose a suitable and easy-to-compute Riemannian metric. But the Dirichlet model is one of the exceptions where the expected Fisher information can be computed exactly and efficiently. How likely is it that SGRLD would be applied to a more general area or probably with some extra approximation?

In order to converge to the exact posterior distribution, SGLD has to anneal the step size towards 0, which will slow down the mixing of the Markov chain even with the help of RLD. Is it possible to run SGRLD with a fixed step size for a faster mixing at the price of introducing bias in the stationary distribution? Also, is there any strategy/heuristic to choose appropriate step size parameters instead of doing a grid search?

Among the four parameterization, RLD + expanded-mean is shown to obtain the best performance. However, with the mirroring trick, the log-likelihood of the model is not differentiable at 0. Would that result in extra error of Langevin dynamics?

In Table 1, what does “n.” mean? Also, it seems there are some errors in G(\theta). E.g. in reduced-mean, there is a factor of n, but in expanded-mean, G(\theta) does not depend on n. So is reduced-nature vs. expanded-natural. Besides, \theta in reduced-nature should be replaced with exp(\theta).

The text and markers in all the figures are too small to read. It would be better to put the description of the parameter values in the caption.

Other typos:
Line 42, are based a batch → missing “on”
Line 79, a SGLD → an SGLD
Figure 1(b), Phi → \pi
Line 399, The algorithms’ performances … was assessed
Line 402, in vocabulary size → missing “a”
Summary: A promising development of SGLD is introduced in this paper. The new algorithm provides a scalable MCMC method to do full Bayesian inference for LDA models, and achieves the state of art.
Author Feedback

Author rebuttal: Many thanks for taking the time to review this paper. Please find answers to the questions raised in the reviews below:

Reviewer 4:
"Is preconditioning necessary to achieve better performance? Did you try to use a non-preconditioned version?"
Yes, preconditioning (or rather using the Riemannian geoemtry) is important for the method to work. This is explored in Section 2.

"Why do you take the absolute value as in line 198? What does this mean for the posterior sampling? Is there a theory behind this?"
There is a theory behind this, outlined briefly in Section 2. It can be understood as using a prior over the whole real line consisting of a gamma distribution along positive reals and its reflection along the negative reals (and dividing by two to maintain total probability of 1). This helps in getting rid of boundaries which simplifies the algorithm.

"Can you give some intuitions or explanations why the proposed method is better?"
We believe the main advantage is that it does not make variational approximations, so it can converge to the correct (and better performing) posterior.

Reviewer 5:
"The authors do not cite any prior working using the expanded mean parameterisation, but it seems surprising if this has not been used before anywhere"
We agree this seems obvious in hindsight. To the best of our knowledge it has not been explored.

Reviewer 6:
"The combination of SGLD with RLD provides a promising direction to improve the application of MCMC methods on large scale data set. It is usually difficult to choose a suitable and easy-to-compute Riemannian metric. But the Dirichlet model is one of the exceptions where the expected Fisher information can be computed exactly and efficiently. How likely is it that SGRLD would be applied to a more general area or probably with some extra approximation? "
Interestingly using the obvious parametrisation with expected Fisher information does not work as well (see Section 2). We do expect similar techniques to be applicable, but that is not in the focus of this paper.

"In order to converge to the exact posterior distribution, SGLD has to anneal the step size towards 0, which will slow down the mixing of the Markov chain even with the help of RLD. Is it possible to run SGRLD with a fixed step size for a faster mixing at the price of introducing bias in the stationary distribution? Also, is there any strategy/heuristic to choose appropriate step size parameters instead of doing a grid search?
We believe yes it is possible to run with a fixed step size. We did not thoroughly explore the question of how best to set appropriate step size parameters in this paper. This is an interesting direction in future, but will have to depend importantly on the specific application of interest.

"Among the four parameterization, RLD + expanded-mean is shown to obtain the best performance. However, with the mirroring trick, the log-likelihood of the model is not differentiable at 0. Would that result in extra error of Langevin dynamics?"
With probability one no proposals with a zero component will be made and as such we don't believe this will cause an issue.

"In Table 1, what does “n.” mean? Also, it seems there are some errors in G(\theta). E.g. in reduced-mean, there is a factor of n, but in expanded-mean, G(\theta) does not depend on n. So is reduced-nature vs. expanded-natural. Besides, \theta in reduced-nature should be replaced with exp(\theta). "
n. means the sum of the count vector n, we will clarify this. Factors of n. - yes there should be a 1/n. factor in G for expanded-mean and \theta in reduced-natural should be replaced with \pi. There is no n. factor in G for the expanded-natural parameterisation. This asymmetry results from the form of the Fisher information of the Poisson distribution.

Reviewers 5&6: Figure text too small to read. Yes we will fix in final version if accepted.